# Factors Associated with Missed Opportunities for Vaccination in Children During the First Year of Life: A Cross-Sectional Study

**DOI:** 10.3390/vaccines13111129

**Published:** 2025-11-01

**Authors:** Wágnar Silva Morais Nascimento, Eugênio Barbosa de Melo Júnior, Ana Raisla de Araújo Rodrigues, Beatriz Mourão Pereira, Joaquim Guerra de Oliveira Neto, Paulo de Tarso Moura Borges, Antonio Rosa de Sousa Neto, Telma Maria Evangelista de Araújo

**Affiliations:** Health Sciences Center, Federal University of Piauí, Teresina 64049-550, Brazil; eugeniomelo@ufpi.edu.br (E.B.d.M.J.); anaraisla@ufpi.edu.br (A.R.d.A.R.); beatriz.pereira@ufpi.edu.br (B.M.P.); joaquim.oliveira@ufpi.edu.br (J.G.d.O.N.); pauloborges@ufpi.edu.br (P.d.T.M.B.); antonioneto@ufpi.edu.br (A.R.d.S.N.); telmaevangelista@ufpi.edu.br (T.M.E.d.A.)

**Keywords:** vaccination coverage, missed opportunity, child health, primary health care

## Abstract

Background: Addressing Missed Opportunities for Vaccination (MOV) contributes to increased vaccination rates in children, reinforcing the need to investigate and intervene in the related factors. Objective: To analyze factors associated with missed opportunities for vaccination in children under one year of age in a Brazilian capital. Methods: This was a cross-sectional, analytical study conducted in seven Basic Health Units in Teresina, Piauí, Brazil. A previously validated questionnaire was applied to parents or guardians of a sample of 316 children. Data were collected from March to June 2025. Multivariable Logistic Regression was performed, and results were expressed as Odds Ratios. Results: Among the children, 53.5% had at least one MOV. The associated factors were: parents with two or more children (95% CI: 1.06–2.96), false contraindications (95% CI: 1.29–8.73), inadequate assessment of vaccination cards by health professionals (95% CI: 1.78–29.00), vaccine shortages in health units (95% CI: 1.57–18.28), and refusal to open multidose vaccine vials (95% CI: 1.81–19.31). Receiving information about vaccination in the previous month was a protective factor against MOV (95% CI: 0.25–0.77). The vaccines most frequently contributing to MOV were BCG (15.8%) and the COVID-19 vaccine, with 15.5% for the first dose and 14.9% for the second. Conclusions: The high prevalence of MOV found in this study indicates weaknesses in the immunization process and suggests the need for implementing measures to interrupt the chain of causes leading to MOV, thereby contributing to the achievement of the objectives of the Brazilian National Immunization Program.

## 1. Introduction

Vaccination is among the most effective public health interventions to reduce morbidity and mortality from infectious diseases. It is a cost-effective primary prevention strategy with substantial benefits for population health, particularly among vulnerable groups such as children in their first year of life [1,2].

Low vaccination coverage during this period poses a considerable risk of morbidity and mortality from vaccine-preventable diseases such as poliomyelitis, measles, pertussis, and hepatitis B. Infants are especially susceptible, as complications from respiratory and gastrointestinal infections tend to be more severe in this age group. Active immunization provides effective protection against many of these diseases and substantially reduces early childhood mortality [3].

A key factor contributing to incomplete vaccination is the occurrence of Missed Opportunities for Vaccination (MOV), defined as situations in which eligible individuals come into contact with health services but do not receive the indicated vaccine doses [4]. Addressing MOV is essential to improving childhood vaccination coverage in both hospitals and primary health care settings [5].

Globally, the prevalence of MOV ranges widely, from 39.8% to 76%, with the highest rates reported in African countries [6,7]. In Latin America, prevalence is lower, ranging between 5% and 37% [8,9]. Multiple factors contribute to MOV, including vaccine shortages, false contraindications, common illnesses in early childhood, institutional shortcomings, and socioeconomic barriers [10]. For example, a review in Latin America highlighted obstacles such as insufficient infrastructure, human resource limitations, and socioeconomic constraints as major challenges [11]. Similarly, a study in Ondo State, Nigeria, found that one in three children experienced MOV during a health care visit [12].

Despite research advances, important gaps remain in strategies to reduce MOV among children under one year of age. In particular, the integration of MOV-prevention measures into vaccination communication programs remains underexplored [13]. Furthermore, recent national studies assessing MOV prevalence are lacking, underscoring the need for new evidence. Identifying the factors associated with MOV is crucial to restoring adequate vaccination coverage and achieving high immunization rates.

This study is warranted given its potential contribution to strengthening vaccination strategies and promoting interventions. It seeks to ensure that children under one year of age have access to all vaccines recommended in the national immunization schedule. It also encourages the development of targeted measures, such as training health professionals to recognize and address MOV and implementing practices that facilitate caregivers’ access to immunization whenever children engage with health services. From this perspective, the present study aimed to analyze factors associated with missed opportunities for vaccination in children under one year of age in a Brazilian capital.

## 2. Materials and Methods

### 2.1. Study Design and Setting

This cross-sectional, analytical study was conducted in line with the Strengthening the Reporting of Observational Studies in Epidemiology (STROBE) guidelines. The research was carried out in seven randomly selected Basic Health Units (BHUs) located in the northern region of Teresina, capital of the state of Piauí, Brazil. BHUs constitute the main entry point to the Brazilian Unified Health System (Sistema Único de Saúde, SUS), the country’s public health system. They are primary health care centers that play a pivotal role in community health by providing a broad range of promotion, prevention, and treatment services, including medical and nursing consultations, vaccination, dental care, medication dispensing, and laboratory specimen collection. BHUs also deliver prenatal care, monitor chronic conditions such as diabetes and hypertension, perform wound care, and carry out health education activities—including home visits—conducted by the multidisciplinary teams responsible for the population assigned to each unit.

### 2.2. Sample

The study population comprised children in their first year of life residing in Teresina. According to the Brazilian Institute of Geography and Statistics [14], this group totaled 10,872 children, of whom 4062 lived in the northern region of the city. As children could not respond to the survey themselves, parents or legal guardians were interviewed.

The sample size was based on a previous study conducted in Teresina [15], which reported a 29% prevalence of MOV among children under two years of age. Assuming this prevalence, with a 5% margin of error and a 95% confidence level, the required sample size was calculated as 316 parents or guardians. Participants were recruited through convenience sampling from children attending the selected health units for any reason.

Children whose vaccination cards were available at the time of data collection were included. Those whose parents or guardians were unable to answer the questionnaire were excluded.

### 2.3. Study Variables

To assess the adequacy of children’s vaccination status, the 2025 immunization schedule of the Brazilian National Immunization Program (NIP) was used as the reference, covering the period from birth to 12 months of age. At birth, children are scheduled to receive the Bacillus Calmette–Guérin (BCG) vaccine against tuberculosis and the hepatitis B vaccine. At 2 and 4 months, they are scheduled to receive the pentavalent vaccine (diphtheria, tetanus, pertussis, Haemophilus influenzae type b, and hepatitis B), the Inactivated Poliovirus Vaccine (IPV), the 10-valent Pneumococcal Conjugate Vaccine (PCV10), and the rotavirus vaccine. At 3 and 5 months, the Meningococcal C vaccine (MenC) is scheduled. At 6 months, children are scheduled to receive the pentavalent vaccine, IPV, and the pediatric Coronavirus Disease 2019 (COVID-19) vaccine, followed by an additional pediatric COVID-19 dose at 7 months. At 9 months, they are scheduled to receive both the pediatric COVID-19 vaccine and the yellow fever vaccine. Finally, at 12 months, the Measles-Mumps-Rubella (MMR) vaccine is scheduled.

### 2.4. Data Collection and Research Instrument

Data collection took place between March and June 2025, after parents or guardians provided written informed consent.

An adapted version of the “health facility exit interview” instrument developed by the World Health Organization (WHO) [5] for MOV surveys was applied. Because the original WHO instrument was designed in 2018 and new vaccines were later incorporated into the childhood immunization schedule, a content validation process was undertaken. Five expert judges evaluated the instrument for relevance, pertinence, appearance, and comprehensibility. The Content Validity Index (CVI) was calculated, and all items achieved a CVI of 100%. The final instrument contained 50 questions grouped into five dimensions. To ensure the content validity of the instrument, a panel of five expert judges was invited. The selection of judges was based on an analysis of their curricula on the Lattes Platform, prioritizing professionals with relevant experience and scientific production in the study’s thematic area. Accordingly, the expert committee comprised three PhDs and two MScs with experience in vaccination, research, and the validation and adaptation of instruments [16].

The experts were asked to evaluate each item of the instrument according to four criteria: clarity, pertinence, relevance, and representativeness. Agreement among judges was assessed using the Content Validity Index (CVI), calculated as the proportion of experts assigning scores of 3 or 4 to each item. Additionally, all comments and suggestions provided by the judges were analyzed and incorporated into the final version of the instrument, improving its quality and suitability for the target population. A CVI of 100% was achieved. The final instrument, therefore, consisted of 50 questions, grouped into five dimensions.

Although a semantic evaluation was not conducted with the target population, a pilot test was carried out with 32 parents or guardians of children residing in the study area to assess the instrument’s clarity and identify potential difficulties. These participants were not included in the main analysis.

In addition to administering the instrument to parents or guardians, the children’s vaccination cards were also examined. Photographic records were made of the section containing vaccination annotations and schedules, identified by the corresponding instrument code, thereby ensuring the confidentiality of the child’s identity.

### 2.5. Data Analysis

Data were analyzed using RStudio (R version 3.6.0). Descriptive statistics included absolute and relative frequencies, measures of central tendency and dispersion, and calculation of MOV prevalence. The prevalence of MOV was calculated as the ratio between the number of children eligible for vaccination who attended the BHU but were not vaccinated (N) and the total number of children within the target age group who attended the health unit during the study period (T), multiplied by 100, that is: MOV = (N/T) × 100.

To assess the combined effect of predictors on the outcome (MOV), multiple logistic regression (MLR) was employed, with results expressed as odds ratios (OR) and their respective 95% confidence intervals (CI) [17]. Variable selection followed a hybrid approach (both exploratory and theoretical): variables with *p* < 0.20 in the bivariate analysis were considered candidates for the multiple model, in order to avoid the premature exclusion of potentially relevant predictors [18,19]. Candidate variables were then simultaneously entered into the model using the Enter (forced entry) method—a recommended procedure when the main objective is explanatory (estimating adjusted effects of predictors) rather than predictive [20]. The significance level adopted for the final model was 5% (α = 0.05).

Overall model adequacy was verified using the Hosmer–Lemeshow goodness-of-fit test (indicating good fit when *p* > 0.05) and complemented by performance indicators: Nagelkerke’s pseudo-R^2^ = 0.2235 and area under the ROC curve (AUC) = 0.78, demonstrating satisfactory fit and adequate discriminative capacity. Multicollinearity was examined using the Variance Inflation Factor (VIF), adopting a cutoff of VIF > 4 [21]; no variable exceeded this threshold, indicating the absence of problematic correlations among predictors and stability of the estimates.

### 2.6. Ethical Considerations

The study was approved by the Research Ethics Committee of the Federal University of Piauí (protocol number 7.444.719).

## 3. Results

The sample consisted predominantly of children aged 0–6 months (78.5%), with a slight male predominance (52.5%). Most parents or legal guardians were female (87.3%), the child’s mother (84.5%), of mixed race (66.5%), residing in urban areas (91.1%), aged 16–29 years (51.9%), married or in a stable union (71.2%), with a high school education (49.1%), and reporting a household income above one minimum wage (58.5%). In half of the cases, the decision to vaccinate was made by the mother, followed by family consensus (48.4%). Almost all respondents (99.7%) reported no religious beliefs interfering with vaccination (Table 1).

As shown in Table 2, 53.5% of children experienced at least one MOV, and 52.8% had an incomplete vaccination schedule for their age.

Table 3 shows MOV prevalence by vaccine type and dose. The highest proportions were observed for BCG (single dose, 15.8%), followed by COVID-19 D1 (15.5%) and D2 (14.9%). Among multidose vaccines, MenC D1 accounted for 10.1%, followed by IPV D2 (9.2%). The lowest prevalence of MOV was observed for yellow fever (single dose, 3.8%).

Among the reasons for MOV related to health professionals were the lack of guidance about vaccination (24.3%), contraindication due to mild illness in the child (25.4%), incorrect statement that vaccination was not scheduled despite being due (13.6%), refusal to vaccinate because the vial could not be opened (17.2%), and contraindication due to a reaction to the previous dose (1.8%). Factors related to parents or guardians included hesitancy following an adverse reaction to a previous dose (6.5%) and the voluntary decision not to vaccinate the child (33.7%). Regarding health service–related factors, vaccine shortages predominated, accounting for 65.1% of the reported reasons (Table 4).

Of the nine variables statistically associated with MOV in the bivariate analysis, six remained significant in the multivariate model. Having two or more children increased the likelihood of MOV by 68% (95% CI: 1.06–2.96). Exposure to vaccination information in the previous month reduced the odds of MOV by 56% (95% CI: 0.25–0.77). Among health professional–related factors, contraindicating vaccination in the presence of mild illness, incorrectly stating that the date was not scheduled for vaccination, and refusing to open a multidose vial were associated with 3.35-, 7.18-, and 5.92-fold higher odds of MOV, respectively (95% CI: 1.29–8.73; 1.78–29.00; 1.81–19.31). In addition, vaccine shortages increased the odds of MOV by 5.36 (95% CI: 1.57–18.28) (Table 5).

## 4. Discussion

This study revealed a high prevalence of MOV and incomplete vaccination among the children assessed. The main factors associated with higher MOV rates were false contraindications, inadequate review of vaccination cards by health professionals, vaccine shortages, and refusal to open multidose vials. Conversely, receiving information or reminders about vaccination in the previous month was protective, reducing the likelihood of MOV.

The predominance of children aged 0–6 months underscores the importance of early interventions. This stage is marked by heightened parental vigilance, increased vulnerability of infants, and a concentration of vaccine doses within the immunization schedule. Nevertheless, the risk of missed opportunities remains, even with frequent health service visits. Regarding sex distribution, there was a slight male predominance, although in other contexts, female children may predominate.

Childhood vaccination was found to be primarily the responsibility of mothers, reinforcing the central role of women in monitoring children’s health and ensuring adherence to immunization. This highlights the critical role of maternal autonomy in achieving full vaccination coverage. By contrast, male-headed households tend to show more missed opportunities, either due to limited male participation in health care or constraints on women’s decision-making power [22,23,24].

A predominance of urban residents was also observed, with a balanced age distribution between younger and older guardians. This profile is generally associated with higher vaccination coverage, as urban areas typically offer a greater concentration of health services, better geographic access, and more frequent campaigns. In contrast, rural contexts are marked by longer distances, workforce shortages, and socioeconomic inequalities that hinder adherence [24,25]. Evidence also suggests that younger mothers are more likely to follow the vaccination schedule, while older mothers—often managing multiple responsibilities—may demonstrate lower adherence [24].

Another relevant aspect was the predominance of mothers married or in stable unions as primary caregivers, along with a higher proportion of single-child families. These characteristics are usually linked to greater family stability, spousal support, and closer monitoring of child health, which may reduce vaccination delays. In line with this, previous studies indicate that firstborn children are more likely to achieve full vaccination coverage, whereas children from larger families face greater challenges in maintaining complete schedules [22,25].

The associations with education and income reinforce the influence of socioeconomic factors on immunization. Parents with higher education are more likely to understand vaccination schedules and value prevention. Conversely, families in vulnerable situations often encounter financial and logistical barriers that increase the risk of MOV [23,26]. Although not statistically significant in this study, proximity to health facilities and access to transportation generally facilitate adherence, whereas long travel times and poor infrastructure are linked to incomplete vaccination [25].

The fact that 53.5% of children experienced at least one MOV strongly indicates weaknesses in vaccination coverage in this setting. This prevalence is notably higher than that reported in studies from Somalia and sub-Saharan Africa, where 26% and 34% of children, respectively, experienced MOV [23,27].

In Brazil, reported MOV prevalence ranges from 16.4% to 50% [10,26]. The findings of this study therefore indicate a prevalence above the national average and higher than most international reports, suggesting the presence of specific local factors negatively affecting vaccination coverage.

More than half of the children also had incomplete vaccination schedules. This is a cause for concern, as it highlights a major gap in coverage within the study population. A similar prevalence was observed in a study conducted in central Brazil, where 52.8% of children living in quilombola communities and rural settlements had incomplete vaccination schedules [28].

Over half of the children in this study experienced MOV. Of the 487 doses not administered or delayed, BCG accounted for the largest proportion, followed by the COVID-19 vaccine. Regarding BCG, parents reported that many facilities restricted administration to a single day per week, ostensibly to optimize vaccine use.

The high prevalence of MOV related to BCG is concerning, given its crucial role in protecting against severe forms of tuberculosis when administered early in life. In the Gozamen district of Ethiopia, BCG accounted for 17.3% of MOV, second only to the zero dose of oral poliovirus vaccine [27]. In Dschang, Cameroon, BCG MOV was also high (16.47%), surpassing that of other vaccines [28]. These findings highlight persistent operational challenges in BCG delivery, despite its prioritization at birth alongside hepatitis B.

The WHO recommends administering BCG as early as possible, ideally still in the maternity ward. However, this has not been consistently achieved. For example, a study in Londrina, Paraná, Brazil, found that many doses were not administered due to a lack of staff trained in the injection technique [28]. Internationally, in Guinea-Bissau, only 19% of newborns received BCG within the first three days of life, though coverage reached 93% by 12 months. This illustrates how bureaucratic and logistical delays hinder timely vaccine delivery [29,30].

COVID-19 vaccine hesitancy remains a major barrier [31]. Studies indicate that part of the population fears adverse events and mistrusts vaccine production, manufacturers, and safety, often linked to reliance on unqualified or unreliable information sources [30,32,33,34]. Nevertheless, pediatric COVID-19 vaccination—starting at six months of age—has demonstrated safety, robust immunogenicity, and moderate efficacy against variants such as Omicron, with protection estimates ranging from 50% to 73% [35,36]. Improving coverage, therefore, requires evidence-based practices by health professionals and managers to reduce barriers, strengthen public confidence, and protect children under one year of age, who face elevated risk of COVID-19–related mortality [37,38,39].

This study also demonstrated the impact of household size on vaccination outcomes. Families with more children faced a greater risk of MOV, consistent with a population-based study in Rondonópolis, Brazil, where additional siblings tripled the likelihood of incomplete vaccination (OR = 3.18; 95% CI: 1.75–5.76) [39]. A study across four northeastern Brazilian cities similarly found that children with siblings had a higher probability of incomplete vaccination (OR = 1.20; 95% CI: 1.11–1.32) [40]. These results underscore the logistical challenges faced by larger households and highlight the need for active case-finding strategies and integration with community health workers.

Basic knowledge of the vaccination card (e.g., using it to record dates) and recent exposure to vaccine-related information were protective against MOV. These findings align with the Vaccination Coverage Survey in Campinas, Brazil, which identified lack of guidance and deliberate refusal to vaccinate as key barriers to full immunization, especially when caregivers lacked consistent, qualified sources of advice [26].

Another critical factor is the inappropriate contraindication of vaccines by health professionals in the presence of mild conditions (e.g., common cold, low-grade fever, constipation, cough, diarrhea, malnutrition, anemia, dehydration) and the inadequate review of vaccination cards. A study in Quito, Ecuador, showed that 98.2% of 273 health professionals interviewed had inappropriately denied vaccination at least once, revealing widespread misinterpretation of true contraindications [41]. Evidence also shows that MOV rates in Latin America range from 5% to 37%, largely influenced by such misinterpretations [8,9,42].

The refusal to open multidose vials was another major contributor to MOV. Administratively, this practice reflects structural challenges. Some multidose vaccines can only be used for a few hours after reconstitution, even under proper storage, discouraging health professionals from opening them without scheduled demand. This is particularly common with lyophilized vaccines such as MMR and BCG [43,44].

In Brazil, fear of vaccine wastage has frequently led to refusals to open multidose vials, undermining the performance of the National Immunization Program (NIP) and limiting childhood vaccination. Addressing this requires integrated actions, including continuous training of health professionals, accurate identification of true contraindications, and encouragement to administer all indicated vaccines even at the risk of vial wastage.

Finally, vaccine shortages in health facilities strongly contributed to increased MOV. Disruptions in supply compromise vaccination coverage and heighten the risk of reintroducing previously controlled or eliminated diseases. Factors underlying these shortages include poor planning of vaccine needs by staff, insufficient training, limited managerial engagement, and the progressive underfunding of the SUS, all of which undermine the effectiveness of the NIP [45].

Notably, unlike some studies [44,45], MOV in this study was not associated with parental hesitancy but with professional and managerial shortcomings, particularly in vaccination room practices and systemic weaknesses across multiple levels of service delivery.

### Study Limitations

This study’s main limitation is that the sample was not randomly selected, which restricts the generalization of the findings to children attending the selected health units. However, this approach is consistent with methodologies commonly used to analyze MOV, given its definition. In addition, children were recruited while attending BHUs for various procedures—most often after medical consultations—rather than specifically from vaccination rooms. Despite these limitations, the findings remain relevant and provide valuable insights into the underlying reasons for missed opportunities for vaccination.

## 5. Conclusions

The high prevalence of MOV identified in this study reveals significant weaknesses in the immunization process. The high rate of MOV underscores the urgent need for measures to optimize vaccination services and highlights specific barriers present in the study context. The vaccination status of the children assessed also showed important gaps, with a substantial proportion having incomplete schedules, underscoring the need to strengthen routine immunization and improve access to services.

The main contributors to MOV were false contraindications, inadequate review of vaccination cards by health professionals, and structural and organizational barriers within health services. Addressing these issues requires coordinated action across municipal, state, and national levels to ensure integrated, continuous, and effective strategies that expand both access to and adherence to vaccination. Although this study focused on children attending health units, the findings suggest that tackling this problem demands alignment among municipal, state, and national management levels to guarantee comprehensive, sustained, and effective vaccination strategies.

These results underscore the urgency of targeted interventions to restore vaccination coverage, prevent the resurgence of vaccine-preventable diseases, and mitigate long-term public health consequences. Essential measures include ensuring a consistent vaccine supply, strengthening distribution logistics, continuously monitoring stocks, and training health professionals. Moreover, expanding access through strategies such as extended service hours and weekend vaccination could play a decisive role in achieving the objectives and targets set by the NIP.

## Figures and Tables

**Table 1 vaccines-13-01129-t001:** Characterization of the studied sociodemographic variables. Teresina, Piauí, Brazil, 2025. (*n* = 316).

Variables	*n*	%
Child’s age group (months)		
0–6	248	78.5
7–11	68	21.5
Guardian’s age group (years)		
16–29	164	51.9
≥30	152	48.1
Child’s sex		
Female	150	47.5
Male	166	52.5
Guardian’s sex		
Female	276	87.3
Male	40	12.7
Race/skin color		
White	54	17.1
Black	47	14.9
Mixed race	210	66.5
Asian	3	0.9
Indigenous	2	0.6
Area of residence		
Urban	288	91.1
Rural	23	7.3
Peri-urban	5	1.6
Marital status		
Married/stable union	225	71.2
Single/separated	90	28.5
Widow(er)	1	0.3
Relationship with the child		
Mother	267	84.5
Father	39	12.3
Grandparent	8	2.5
Other	2	0.6
Total number of children		
1	172	54.4
≥2	144	45.6
Guardian’s education		
No formal education	2	0.6
Elementary school	25	7.9
High school	155	49.1
Incomplete higher education	42	13.3
Complete higher education	92	29.1
Household income (based on BRL 1518.00)		
Up to 1 minimum wage	131	41.5
More than 1 minimum wage	185	58.5
Travel time to health unit (minutes)		
≤15	239	75.6
>15	77	24.4
Means of transportation to health unit		
On foot	53	16.8
Bicycle	1	0.3
Motorcycle	46	14.6
Private car	175	55.4
Public transport	5	1.6
App-based transport	38	12.0

**Table 2 vaccines-13-01129-t002:** Prevalence of missed opportunities for vaccination and vaccination status per child. Teresina, Piauí, Brazil–2025 (*n* = 316).

Variables	*n*	%	95% CI	
			LL	UL
Missed Opportunities for Vaccination (MOV)				
Yes	169	53.5	48.0	59.0
No	147	46.5	41.0	52.0
Vaccination status				
Complete schedule	149	47.2	41.6	52.7
Incomplete schedule	167	52.8	47.3	58.4

Legend: CI–Confidence Interval; LL–lower limit; UL–upper limit.

**Table 3 vaccines-13-01129-t003:** Prevalence of missed opportunities for vaccination according to vaccine type and dose. Teresina, Piauí, Brazil–2025 (*n* = 169).

Vaccine Type	Dose	MOV		95% CI	
		*n*	(%)	LL (%)	UL (%)
BCG	BCG (single dose)	50	15.8	11.8	19.8
Hepatitis B	Hepatitis B (single dose)	39	12.3	8.7	16.0
IPV	IPV D1	25	7.9	4.9	10.9
	IPV D2	29	9.2	6.0	12.4
	IPV D3	19	6	3.4	8.6
Pentavalent (Penta)	Penta D1	23	7.3	4.4	10.1
	Penta D2	26	8.2	5.2	11.3
	Penta D3	17	5.4	2.9	7.9
Rotavirus	Rotavirus D1	23	7.3	4.4	10.1
	Rotavirus D2	23	7.3	4.4	10.1
Pneumococcal 10-valent (PCV10)	PCV10 D1	24	7.6	4.7	10.5
	PCV10 D2	23	7.3	4.4	10.1
Meningococcal C (MenC)	MenC D1	32	10.1	6.8	13.5
	MenC D2	26	8.2	5.2	11.3
COVID-19	COVID-19 D1	49	15.5	11.5	19.5
	COVID-19 D2	47	14.9	11.0	18.8
Yellow Fever	Yellow fever (single dose)	12	3.8	1.7	5.9

Legend: CI–Confidence Interval; LL–lower limit; UL–upper limit; D1–dose 1 (first dose); D2–dose 2 (second dose); D3–dose 3 (third dose).

**Table 4 vaccines-13-01129-t004:** Distribution of missed opportunities for vaccination according to reasons related to health professionals, parents, and vaccination services. Teresina, Piauí, Brazil–2025 (*n* = 169).

Variables	MOV		95% CI	
	*n*	%	LL	UL
Reasons related to health professionals				
The health professional (physician or nurse) provided information about vaccination				
Provided information	128	75.7	71.0	80.5
Did not provide information	41	24.3	19.5	29.0
Health professional contraindicated vaccination due to mild illness in the child				
Yes (Inappropriate contraindication)	43	25.4	20.6	30.2
No (Did not contraindicate vaccination)	122	72.2	67.2	77.1
Did not remember	4	2.4	0.7	4.0
Professional incorrectly stated that the child was not scheduled for vaccination				
Yes (Incorrect assessment of the vaccination card)	23	13.6	9.8	17.4
No (Correct assessment of the vaccination card)	146	86.4	82.6	90.2
Vaccination professional refused to vaccinate the child claiming the vial could not be opened				
Failed to vaccinate for this reason	29	17.2	13.0	21.3
Did not fail to vaccinate for this reason	140	82.8	78.7	87.0
Health professional contraindicated vaccination because the child had a reaction to the previous dose				
Contraindicated the vaccine	3	1.8	0.3	3.2
Did not contraindicate the vaccine	166	98.2	96.8	99.7
Reasons related to parents				
Considered not vaccinating again because the child had a reaction to a previous dose				
Considered not vaccinating the child	11	6.5	3.8	9.2
Did not consider not vaccinating the child	158	93.5	90.8	96.2
Failed to vaccinate the child on the scheduled date by their own choice				
Failed to vaccinate	57	33.7	28.5	38.9
Did not fail to vaccinate	112	66.3	61.1	71.5
Reasons related to vaccination services				
Failed to vaccinate because:				
Vaccines were out of stock	110	65.1	59.8	70.3
Syringes or other vaccination supplies were unavailable	28	16.6	12.5	20.7
It was not a vaccination day at the health unit	12	7.1	4.3	9.9
The vaccination room was closed	44	26.0	21.2	30.9
The vaccination staff member was absent	27	16.0	11.9	20.0
Waiting time was too long	20	11.8	8.3	15.4
Staff or team member was disrespectful	7	4.1	1.9	6.3
Vaccination room opening hours were inadequate	11	6.5	3.8	9.2
Vaccination room operated in a single shift	21	12.4	8.8	16.1

Legend: CI–Confidence Interval; LL–Lower limit; UL–Upper limit.

**Table 5 vaccines-13-01129-t005:** Multiple logistic regression of factors associated with missed opportunities for vaccination. Teresina, Piauí, Brazil–2025 (*n* = 169).

Variables	Bivariate	Multivariate
	OR ᵇ (95% CI)	OR ᵃ (95% CI)
Child’s age group (months)		
0–6 months	1	1
7–11 months	2.52 [1.41–4.49] *	1.75 [0.86–3.55]
Child’s sex		
Female	1	1
Male	1.37 [0.88–2.14] *	1.55 [0.9–2.69]
Number of children		
1	1	
≥2	1.59 [1.02–2.49] *	1.68 [1.06–2.96] **
Knows the purpose of the vaccination card		
Yes	0.25 [0.05–1.15] *	0.22 [0.04–1.12]
No	1	
Believes all recommended vaccines are necessary		
Yes	0.3 [0.08–1.09] *	0.35 [0.08–1.54]
No	1	
Received information about vaccination in the last month		
Yes	2.14 [1.36–3.38] *	0.44 [0.25–0.77] **
No	1	
Main sources of information about vaccination		
Health professionals	0.64 [0.33–1.21] *	0.69 [0.35–1.36]
Internet/TV/books/vaccination card	1.46 [0.75–2.88]	1.58 [0.78–3.23]
Relatives and friends	0.58 [0.21–1.57]	0.72 [0.25–2.03]
Health professionals contraindicated vaccination due to mild illness		
Yes	6.9 [2.99–15.9] *	3.35 [1.29–8.73] **
No	1	
Professional stated it was not the vaccination date (even though it was)		
Yes	7.56 [2.22–25.74] *	7.18 [1.78–29] **
No	1	
Professional refused vaccination claiming vial could not be opened		
Yes	7.41 [2.54–21.61] *	5.92 [1.81–19.31] **
No	1	
Parents considered not vaccinating after reaction to previous dose		
Yes	3.34 [0.91–12.22] *	3.09 [0.64–14.78]
No	1	
Parents failed to vaccinate on scheduled date by own choice		
Yes	2.89 [1.66–5.03] *	1.89 [0.97–3.68]
No	1	
Failed to vaccinate your child due to (Service-related reasons)		
Vaccines out of stock	4.43 [1.48–13.2] *	5.36 [1.57–18.28] **
Lack of supplies	0.56 [0.14–2.15]	0.43 [0.10–1.91]
Not a vaccination day at health unit	1.10 [0.19–6.41]	0.81 [0.10–6.36]
Vaccination room closed	0.42 [0.12–1.50] *	0.27 [0.06–1.14]
Vaccination staff absent	1.11 [0.26–4.63]	1.14 [0.22–5.86]
Long waiting time	2.31 [0.50–10.80]	1.86 [0.35–9.78]
Vaccination room operated in a single shift	0.23 [0.03–2.05] *	0.15 [0.01–1.67]

Legend: OR ^b^–Odds Ratio Gross; OR ^a^–Adjusted Odds Ratio; CI–Confidence Interval; * *p* < 0.20; ** *p* < 0.05.

## Data Availability

The data presented in this study are available on request from the corresponding author due to ethical reasons and privacy data.

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
