# Peer review of "Factors Associated with Missed Opportunities for Vaccination in Children During the First Year of Life: A Cross-Sectional Study"

_vaccines, 2025, doi:10.3390/vaccines13111129_

Round 1
Reviewer 1 Report
Comments and Suggestions for Authors
This is an excellent and straightforward study of missed vaccination opportunities in very young children. The introduction, study plan, patient data collection using a validated instrument and data analysis are all excellent.
To guide ministries of health and vaccination programme leaders and participants could you make a box highlighting the key conclusions and offering suggestions about how to decrease each of the key MOV's based on your own clinic experience and discussions with knowledgeable individuals in the vaccination programme?
Author Response
Dear Editors and Reviewers,
We would like to express our gratitude for the opportunity to revise our manuscript for publication in this prestigious journal. We have carefully read all the comments and suggestions and have incorporated almost all of them.
We emphasize that justifications have been provided for the few suggestions not implemented and that all corrections have been highlighted in yellow in the manuscript to facilitate the review.
REVIEWER 1:
To guide the Ministry of Health and the leaders and participants of the vaccination program, could you create a table highlighting the main conclusions and offering suggestions on how to reduce each of the main MOVs, based on your own clinical experience and discussions with individuals experienced in the vaccination program?
RESPONSE:
Thank you for your comment. We agree that strategies to reduce MOVs are crucial; however, creating a detailed table as suggested, although a highly relevant recommendation, goes beyond the scope and objectives of our manuscript.
Reviewer 2 Report
Comments and Suggestions for Authors
Please see my review comments in the attached word doc. Your team is to be congratulated for an over all important study and look forward to revisions
Review of article for Vaccines- vaccines-3915692
10.10.2025
Title: Factors Associated with Missed Opportunities for Vaccination in Children During the First Year of Life: A Cross-Sectional Study
Summary.
This is a well written and highly relevant clinical service project from Brazil. The research group are a group of Public Health investigators that are addressing an important but often unappreciated clinical service problem of missed opportunity in delivering vaccines to young children. This issue is also relevant to all ages. Considering the sentiment within countries particularly the US and the global importance of vaccination for children’s health, this study as described in the submitted manuscript is both timely and highly relevant.
The authors clearly state the importance of vaccination for health with available published work and build support for the stated aims which is stated in clear a logical fashion. The aims of their project were to use well accepted clinical research procedures and gather data in a sample of caregivers of children related to their primary outcome of missed opportunities. They clearly state the many reasons for a missed opportunity and identify the gap in understanding the importance of missed opportunity and the lack of integrating a plan to identify and decrease missed opportunities. They overall developed an implicit testable hypothesis based on their background and the identified unmet needs.
The manuscript is highly structured and is relevant within the field of public health. The strength of the study as stated above is the clinical and practical relevance to clinical care delivery.
I will briefly address each section in the paper.
Study Design Setting, Sample
The design is stated and appropriate and following the STROBE guidelines sampling carried out in seven randomized health units. They conducted a sample calculation based on previous study calculating an effect size and reasonable study sample.
Sampling Validity and Representativeness
The study defines its target population well—children in their first year of life living in Teresina—a municipality in a very large county- Brazil and the sampling strategy raises concerns about representativeness. Participants were recruited through convenience sampling of parents or guardians attending health units, which means not all eligible families had an equal chance of inclusion. Although the sample-size calculation was carefully performed, it assumes random sampling and therefore doesn’t apply accurately to a convenience-based design. This limits how far the findings can be generalized beyond families who visit these facilities. That said, the authors did an excellent job of clearly describing their inclusion and exclusion criteria and were transparent about their data collection procedures. If they simply clarify that their findings reflect health-unit attendees rather than the city’s entire infant population, the conclusions will better align with the study design.
Instrument Validation (WHO Tool Adaptation)
The authors adapted the original WHO instrument given with updates to the national vaccination schedule. However, the validation effort as described appears limited in scope. Only five expert judges were involved in reviewing the instrument for relevance, pertinence, appearance, and comprehensibility, and while a Content Validity Index (CVI) of 100% was reported, this result alone is not sufficient evidence of robust validation. Typically, a more comprehensive process would include cognitive interviews with end-users, pilot testing with the target population, or evaluation of inter-rater reliability. The authors should clarify whether such additional steps were performed and discuss how the expert panel was selected and whether it represented multiple professional backgrounds. That said, the decision to revisit and adapt the WHO instrument was entirely appropriate, and the authors’ transparency in reporting the CVI process and final item structure is appreciated. With further explanation of how feedback was incorporated and how the revised items were tested, this section could more convincingly demonstrate the instrument’s content validity.
Statistical Analysis and Model Validity
The analysis is well executed and clearly described. The authors used RStudio and have used appropriate descriptive and inferential methods. The logistic regression model is a valid choice to explore predictors of missed opportunities for vaccination. However, the definition of MOV prevalence—based only on children attending health units—reflects a local facility-based measure, not a community-wide or population based prevalence, and this distinction should be emphasized. The variable-selection process (p < 0.20 in bivariate analysis followed by an “Enter” model) could use more explanation, since it mixes exploratory and theory-based approaches. Adding a few complementary model-fit indicators (e.g., pseudo-R², AUC) would strengthen confidence in the findings. Overall, the analytic framework is transparent and well executed, but interpretation should be limited to the studied subgroup. The authors’ clear reporting of diagnostics such as VIF and Hosmer–Lemeshow are definite strengths but maybe lost on a general clinician without a strong stats background.
Methods Appraisal
The Methods section is well organized, clearly written, and easy to follow. The authors deserve credit for providing transparent descriptions of their sample-size rationale, inclusion criteria, and analytic process. The main methodological limitations relate to the non-probability sampling and to the interpretation of the MOV prevalence, which should be explicitly identified as facility based. The regression analysis is generally appropriate, though the combination of p-value preselection and full entry requires a brief justification. Despite these limitations, the paper reflects thoughtful design, sound analysis, and commendable reporting transparency.
Results
The section is well constructed and can be reviewed without major issues. The findings in general are very interesting and relevant but are again limited to a facility or regional context. However, this does not limit the importance or generalizability of the study. As a research clinician I believe the areas of need discovered by the authors are highly applicable to most medical care delivery pathways globally.
I will provide comments based on each reported data set which may be helpful to the authors
Table 1- How do these regional demographics compare to the larger Brazil population > 200 million to establish representativeness?
Table 2- Striking troubling outcomes in MOV. Is there any correlative data on the disease outcomes for these kids? That is does the MOVs relate directly to an increase in the various infectious disease?
Table 3- Needs a bit of reorganization and editing to make more readable. You may just report vaccine type and provide dose details in an appendix? Is there any relevance to listing all the sub types of vaccines? Is a subtype a determinant of MOV?
Table 4- The most important data set with Table 5 in my review. This statement was taken from the manuscript… “The main reasons for MOV were lack of information provided by health professionals (75.7%)”. The data reported is a bit confusing as it’s a Yes- No response.
|
Reasons related to health professionals |
||||
|
The health professional (physician or nurse) provided information about vaccination |
||||
|
Yes |
128 |
75.7 |
71.0 |
80.5 |
|
No |
41 |
24.3 |
19.5 |
29.0 |
It may appear that the YES response means the health professional provided information that led to the MOV? Perhaps some clarifying text
Table 5- Lots of data and it will take a dedicated and fastidious general reader to spend a bit of time going through the numbers. You could eliminate displaying some of the categories and just mention in the text or place in a separate section. Also maybe change the reporting format to the OR CI instead of two columns for LL and UL.
Discussion/ Conclusion
Well written and addresses most major issues balancing with published studies. The authors identify the major gaps in the delivery of vaccines to the children and offer potential solutions. Most of organized medicine has now or will soon adapt using EHR/ EMR systems such as EPIC.
It’s plausible and even likely that well-implemented EHR decision support (e.g. EPIC alerts, reminders, IIS linkage) could reduce MOV in infant immunization programs. If the Brazilian study is not already using or considering such digital supports, I would like to know if any health units in Teresina use electronic health records or digital decision support, and how that might affect MOV. I would suggest that the authors discuss existing evidence on EHR-based interventions in immunization and consider whether analogous strategies (alerts, interoperability, reminders) might be applicable in the Brazilian context.
I understand the feasibility of translating EHR-based interventions from high-resource settings (e.g. U.S. with EPIC infrastructure) to settings with variable digital infrastructure which could be addressed.
Overall Assessment
This is a carefully prepared and clearly written manuscript addressing an important public health issue—missed opportunities for vaccination among infants. The authors demonstrate strong understanding of the topic, transparent reporting, and technical competence. The main areas that need improvement involve clarifying the limits of the sampling approach, expanding on the instrument validation process, and refining interpretation of the analytic results. These are methodological and interpretive issues rather than flaws in design or execution. With these clarifications and a more nuanced discussion of limitations, this study could make a valuable contribution to the Brazilian and broader immunization literature.
Recommendation: Major Revision
This paper is well structured, clearly written, and addresses an important question. However, the validity of the results is constrained by the convenience sampling design and by limited detail regarding the instrument’s validation process. These issues are correctable with clarification and do not require new data collection. The manuscript would benefit from expanded explanation of how the WHO tool was validated, acknowledgment that the findings are facility-based, and clearer justification of analytic decisions. I recommend major revision, as the work has significant merit and should be strengthened rather than rejected.
Author Response
Dear Editors and Reviewers,
We would like to express our gratitude for the opportunity to revise our manuscript for publication in this prestigious journal. We have carefully read all the comments and suggestions and have incorporated almost all of them.
We emphasize that justifications have been provided for the few suggestions not implemented and that all corrections have been highlighted in yellow in the manuscript to facilitate the review.
REVIEWER 2:
- Validity and representativeness of the sample
Participants were recruited through convenience sampling of parents or guardians attending health units, which means that not all eligible families had the same chance of inclusion. Although the sample size calculation was carefully performed, it assumes random sampling and therefore does not accurately apply to a convenience-based design. This limits the generalization of the results beyond families who attend these units. That said, the authors did an excellent job clearly describing their inclusion and exclusion criteria and were transparent about their data collection procedures. If they simply clarify that their results reflect those who attend health units rather than the entire child population of the city, the conclusions will align better with the study design.
RESPONSE:
Thank you for your comments. We agree and have added in the study limitations that the use of convenience sampling restricts the generalization of the findings to children attending the selected health units.
- Instrument validation (Adaptation of WHO tools)
The authors adapted the original WHO instrument, incorporating updates for the national immunization schedule. However, the validation effort, as described, appears limited in scope. Only five expert judges were involved in reviewing the instrument for relevance, pertinence, appearance, and comprehensibility, and although a Content Validity Index (CVI) of 100% was reported, this result alone is not sufficient evidence of robust validation. Typically, a more comprehensive process would include cognitive interviews with end users, pilot testing with the target population, or inter-rater reliability assessment. The authors should clarify whether such additional steps were conducted and discuss how the expert panel was selected and whether it represented multiple professional backgrounds. That said, the decision to revisit and adapt the WHO instrument was entirely appropriate, and the authors’ transparency in reporting the CVI process and the final item structure is appreciated. With further clarification on how feedback was incorporated and how the revised items were tested, this section could more convincingly demonstrate the content validity of the instrument.
RESPONSE:
We fully agree and acknowledge that although this important step (pilot testing) of the study was conducted, it was indeed omitted in the initial version. We have now included this information in the instrument validation section and added details about the selection of the expert panel, as well as the information regarding the photographic documentation of vaccination cards, which had not been previously included in the manuscript.
- Statistical analysis and model validity
The analysis is well executed and clearly described. The authors used RStudio and appropriate descriptive and inferential methods. The logistic regression model is a valid choice to explore predictors of missed opportunities for vaccination (MOV). However, the definition of MOV prevalence — based solely on children attending health units — reflects a facility-based measure rather than a community- or population-based prevalence, and this distinction should be emphasized. The variable selection process (p < 0.20 in the bivariate analysis followed by an “Enter” model) may require further clarification, as it mixes exploratory and theory-based approaches. Adding a few complementary indicators of model fit (e.g., pseudo-R², AUC) would strengthen confidence in the results. Overall, the analytical framework is transparent and well implemented, but interpretation should be limited to the subgroup studied. The authors’ clear reporting of diagnostics such as VIF and Hosmer–Lemeshow tests are definite strengths but might be overlooked by general clinicians without a solid statistical background.
RESPONSE:
We agree that the approach adopted required further detail, and this has been added to the manuscript. Following the recommendation, additional indicators of model fit and performance have been included, such as Nagelkerke’s pseudo-R² and the area under the ROC curve (AUC), to reinforce the assessment of goodness-of-fit and discriminative ability.
In addition, a brief explanatory note on the meaning of the diagnostic tests (Hosmer–Lemeshow and VIF) has been inserted to make the text more accessible to readers without advanced statistical training.
- Evaluation of methods
The main methodological limitations relate to the non-probabilistic sampling and the interpretation of MOV prevalence, which should be explicitly identified as facility-based. The regression analysis is generally appropriate, although combining p-value preselection and the full-entry method requires a brief justification. Despite these limitations, the article reflects careful design, solid analysis, and commendable transparency in reporting.
RESPONSE:
The study used non-probabilistic convenience sampling, as participants were recruited from among children attending the selected health units. Only the units themselves were randomly chosen, which limits the generalization of the findings to other populations. This information was reinforced in the study limitations section, emphasizing that the prevalence of missed opportunities for vaccination (MOV) should be interpreted as a facility-based prevalence.
Regarding the combination of variable preselection with p < 0.20 in the bivariate analysis and the use of the Enter method in multiple regression, we clarified that there is no methodological contradiction, as these are complementary steps in the analytical process. However, following your recommendation, we provided a clearer explanation of this in the manuscript.
- Results
I will provide comments based on each dataset reported that may be useful to the authors.
Table 1 – How do these regional demographic data compare to the larger Brazilian population (>200 million) to establish representativeness?
RESPONSE:
Thank you for your comment. We understand the importance of contextualizing our findings in relation to the broader Brazilian population. However, the main objective of this study is not statistical generalization to the national population but rather a detailed and in-depth analysis of the specific population selected for this study.
Table 2 – The MOV results are concerning and striking. Are there any correlational data on disease outcomes in these children? In other words, are MOVs directly related to an increase in various infectious diseases?
RESPONSE:
Your comment is very interesting and pertinent. However, there are no data directly linking MOVs to an increase in various infectious diseases. At present, new cases of pertussis have begun to appear in the state of Piauí, which certainly reflect inadequate coverage of the pentavalent vaccine. However, these are recent cases that emerged after the data collection period of this study.
Table 3 – This table needs some reorganization and editing to improve readability. You could simply indicate the type of vaccine and provide details about the dose in an appendix. Is there any relevance to listing all vaccine subtypes? Is a subtype a determinant of the MOV value?
RESPONSE:
Although your comment is important, we believe that indicating not only the type of vaccine but also the specific dose that has been most frequently missed provides valuable information. It highlights potential vaccine wastage, since vaccines with sequential doses do not have the same effectiveness if only partially administered, and also serves as an alert for health professionals to develop strategies aimed at ensuring completion of the vaccination schedule.
Table 4 – The dataset presented in Table 5 (in my review) appears to be the most important. The statement was taken from the manuscript: “The main reasons for MOV were the lack of information provided by health professionals (75.7%).” The reported data seem somewhat confusing, as the response format is “yes” or “no.”
|
Reasons related to health professionals |
||||
|
Health professional (physician or nurse) provided information about vaccination |
||||
|
Yes |
128 |
75.7 |
71.0 |
80.5 |
|
Not |
41 |
24.3 |
19.5 |
29.0 |
It may appear that the answer “Yes” means that the health professional provided information that led to the MOV. Perhaps some clarifying text would help.
RESPONSE:
We appreciate the observation. Indeed, the wording of the item “The health professional provided information about vaccination” could lead to the interpretation that the percentage of 75.7% (“Yes”) represented a reason for MOV, whereas in reality, this value indicates that guidance was received. For the purposes of this analysis, the reason for MOV is the absence of guidance; therefore, the correct value to be reported as the reason is “Did not provide information” = 24.3%.
To eliminate this ambiguity:
- We have reformulated the text in the Results section to report, for each item, the category that characterizes the reason.
- We have adjusted Table 4 to make it explicit that the percentages presented correspond to the presence of the reason.
Table 5 – There are many data points, and a dedicated and meticulous general reader will need to spend some time examining the numbers. You could remove the display of some categories and simply mention them in the text or place them in a separate section. It might also be possible to change the reporting format to show the CI of the OR instead of two separate columns for the lower and upper limits (LL and UL).
RESPONSE:
We have revised the manuscript and incorporated the suggested changes related to this table, as requested. We chose to remove the p-value from the table to make it cleaner and to avoid redundancy, since the 95% CI already conveys the information on statistical significance.
- Discussion/Conclusion
Well written and addresses most of the key issues, in balance with published studies. The authors identify the main gaps in childhood vaccine administration and offer possible solutions. Most organized healthcare systems have already adapted or are in the process of adapting by implementing electronic health record (EHR/EMR) systems such as EPIC.
It is plausible, and even likely, that decision support through well-implemented electronic records (e.g., EPIC alerts, reminders, linkage with IIS) could reduce value variance in childhood immunization programs. If the Brazilian study is not yet using or considering such digital supports, I would like to know whether any health units in Teresina use electronic health records or digital decision support, and how this might affect value variance. I suggest that the authors discuss existing evidence on EHR-based interventions in immunization and consider whether analogous strategies (alerts, interoperability, reminders) might be applicable in the Brazilian context.
RESPONSE:
Thank you for this highly relevant comment. Although Teresina implemented the Citizen’s Electronic Health Record (Prontuário Eletrônico do Cidadão – PEC) eight years ago, some challenges to achieving better vaccination coverage persist due to inadequate interoperability between the e-SUS system and the National Immunization Information System (SI-PNI).
Reviewer 3 Report
Comments and Suggestions for Authors
The investigators report an ERC-approved cross-sectional survey of parents of infants (n=315) residing in Teresina, Brazil to estimate the prevalence of infant missed opportunities to vaccinate and factors associated with MOVs. Data were sourced from parent interviews and vaccination cards. They found that half of infants had an MOV, that BCG and COVID-19 vaccines were the most common vaccines with MOVs, that factors associated with MOVs included not reading the vaccination card, false contraindications, vaccine stockouts, and that receiving a visit reminder was associated with fewer MOVs. They concluded that “the high prevalence of MOV found in this study indicates weaknesses in the immunization process and suggests the need for implementing measures to interrupt the chain of causes leading to MOV.”
Missed opportunities to vaccinate have been the object of research for decades, and for good reason. Reducing MOVs is known to increase vaccination coverage. Thus the topic of their study is important. The manuscript is clearly written. Their conclusion is based on the data presented.
I have a few suggestions to improve the manuscript.
The sampling strategy is not specified in sufficient detail for the reader to understand how the subjects were selected. (1) The sample was recruited in Basic Health Units. Many readers will not know what BHUs provide. Are they primary care providers, immunization providers, emergency care providers? (2) The reader is told that BHUs were randomly selected, but how were the parents recruited? Were they recruited when bringing their infant to the BHU? If so, was this a clinic-intercept study, and if so, were the parents recruited before or after the BHU provider saw the infant? Alternatively, were parents recruited by sampling from a list of infants served by the BHU? (3) Is it possible to determine a response rate for the parents? If not, then this should be described as a convenience sample. If so, the calculation should be provided.
A major limitation of the study is that there apparently was no medical chart review – other than parent-held vaccination records. Since this is an MOV study, a key aspect of an MOV is that it is a visit to a health care facility in which a child is due for one or more vaccinations, has no contraindication to a due vaccination, and is not provided the due vaccination. Since there is no chart review, only parents’ recall provides information about any MOV. Without record review, one does not know if the provider documented that the child was due for vaccination and the reasons for non-vaccination. Parents are not necessarily the best source of such information. The lack of medical chart review should be indicated in the study limitations section.
Related to the lack of record review, the reader is not told the nature of the visit that resulted in an MOV. Was the MOV visit specifically for vaccination? Was the MOV visit a health supervision visit? Was the MOV visit an illness visit? Was the MOV visit a chronic medical condition visit? Was the MOV visit a follow-up visit? Was the MOV reason documented in the MOV visit? Was a follow-up appointment made to resolve the MOV after an MOV visit? The reader would also benefit from knowing the prevalence of MOVs by visit type.
A related limitation is that (assuming this is a clinic-intercept study) it is not clear if the visit at which parents were recruited was, itself, an MOV. This would be useful for documenting the prevalence of MOVs.
It would be useful to know the distribution of the number of MOV visits for the 315 infant subjects. I assume that some had more than one MOV.
Reference 5 is a reference to the instrument that was modified by the investigators to be used as the instrument for the present study. The instrument in reference 5 has a parent section and a provider section. However, the manuscript does not appear to have results from a provider interview part. Were providers interviewed? If so, the reader would benefit from seeing the results of the provider interviews. It would also be good if the actual instrument for the study could be provided in a supplemental.
A significant point of confusion in the study is the definition of an MOV. The authors’ definition is of MOV prevalence is “MOV prevalence was defined as the ratio between the number of unvaccinated children (N) and the total number of children in the target age group who attended the health units during the study period (T), multiplied by 100: MOV = (N/T) × 100.” Dividing the number of unvaccinated children by the number of children attending the clinic seems more like a definition of undervaccination rate rather than MOV prevalence. An MOV by WHO definition involved a facility visit, but the authors’ definition does not seem to include visits to the facility. The authors should clarify their definition and show how facility visits are involved with the definition of an MOV.
Abstract, methods – The age of the study children should be mentioned.
Author Response
Dear Editors and Reviewers,
We would like to express our gratitude for the opportunity to revise our manuscript for publication in this prestigious journal. We have carefully read all the comments and suggestions and have incorporated almost all of them.
We emphasize that justifications have been provided for the few suggestions not implemented and that all corrections have been highlighted in yellow in the manuscript to facilitate the review.
REVIEWER 3:
I have a few suggestions to improve the manuscript.
The sampling strategy is not specified in sufficient detail for the reader to understand how the subjects were selected.
- The sample was recruited in Basic Health Units (UBSs). Many readers may not know what UBSs provide. Are they primary care providers, immunization providers, emergency care providers?
RESPONSE:
Suggestion addressed. Information describing the activities carried out in the UBSs has been incorporated into the text, in the Study Design and Setting section.
- The reader is informed that the UBSs were randomly selected, but how were the parents recruited? Were they recruited when bringing their babies to the UBS? If so, was this a clinical intercept study, and were the parents recruited before or after the UBS provider saw the baby?
RESPONSE:
We had already specified in the manuscript that the children were recruited during their visit to the UBS, as they exited the consultation rooms after receiving any type of care—such as pediatrics, child health follow-up, or after leaving the vaccination room—that is, after the appointment.
- Alternatively, were the parents recruited by sampling from a list of babies served by the UBS? Is it possible to determine a response rate for the parents? If not, this should be described as a convenience sample. If so, the calculation should be provided.
RESPONSE:
The UBSs were randomly selected; however, the children were recruited through a convenience sample. This is described in the Methods section and is also presented as a limitation of the study, since the findings cannot be generalized to other populations.
- An important limitation of the study is that there was apparently no review of medical records—apart from the vaccination records kept by parents. As this is a study on MOV, a key aspect of an MOV is that it involves a visit to a health unit during which a child should receive one or more vaccines, has no contraindication to vaccination, and does not receive the due vaccination. Without medical record review, only parents’ recollection provides information about any MOV. Without reviewing the records, it is unclear whether the provider documented that the child was due for vaccination and the reasons for non-vaccination. Parents are not necessarily the best source of such information. The lack of medical record review should be indicated in the study’s limitations section.
RESPONSE:
Thank you for your comment. Although it was already stated in the Methods section (Sampling) that having the vaccination card was an inclusion criterion, it was not sufficiently clear that we analyzed and photographed all vaccination cards. Therefore, we revised the text to make it explicit that the data on vaccine doses received were obtained directly from the vaccination cards, not from the parents’ verbal reports.
- Regarding the lack of medical record review, the reader is not informed about the nature of the visit that resulted in an MOV. Was the MOV visit specifically for vaccination? Was it a health supervision visit? Was it a sick visit? Was it a visit for a chronic medical condition? Was it a follow-up visit? Was the reason for the MOV documented during that visit? Was a follow-up appointment scheduled to resolve the MOV after the MOV visit? The reader would also benefit from knowing the prevalence of MOVs by type of visit.
RESPONSE:
The visits were not specifically for vaccination but, as described in the Methods section, could involve any type of care provided at the UBSs: consultations for growth and development assessment with a nutritionist, pediatric consultations with physicians, child health (puericulture) consultations with nurses, or vaccination appointments. None of the children in the study had chronic conditions.
- A related limitation is that (assuming this is a clinical intercept study) it is not clear whether the visit during which the parents were recruited was itself an MOV. Documenting the prevalence of MOVs would be helpful.
RESPONSE:
This information was collected and is presented in Tables 4 and 5. In addition, the Discussion section already addresses the inadequate assessment of vaccination cards, which itself constituted an MOV.
- It would be useful to know the distribution of the number of MOV visits among the 315 participating infants. I assume that some had more than one MOV.
Reference 5 refers to the instrument that was modified by the researchers for use in the present study. The instrument in reference 5 includes a section for parents and a section for providers. However, the manuscript does not appear to present results from a provider interview component. Were providers interviewed? If so, the reader would benefit from seeing the results of those provider interviews. It would also be interesting if the actual study instrument could be provided as a supplement.
RESPONSE:
The providers were not interviewed. We agree that this would have been very useful; however, we chose to focus on a particularly important segment involved in MOVs — the parents of the children.
- A significant point of confusion in the study concerns the definition of an MOV. The authors’ definition of MOV prevalence is: “The prevalence of MOV was defined as the ratio between the number of unvaccinated children (N) and the total number of children in the target age group who attended the health units during the study period (T), multiplied by 100: MOV = (N/T) × 100.”
Dividing the number of unvaccinated children by the number of children who attended the clinic seems more like a definition of under-vaccination rate than of MOV prevalence. According to the WHO definition, an MOV involves a visit to a health unit, but the authors’ definition does not appear to include such visits. The authors should clarify their definition and show how health unit visits are incorporated into the definition of an MOV.
RESPONSE:
Thank you for drawing our attention to this point. It was simply a typographical error that went unnoticed and has now been corrected. The calculation is now properly defined as follows: the ratio between the number of children eligible for vaccination who attended the Basic Health Unit (UBS) and were not vaccinated (N), and the total number of children in the target age group who attended the health units during the study period (T), multiplied by 100: MOV = (N/T) × 100.
This MOV prevalence was calculated using the formula recommended by the WHO, as presented in the study by Hutchins, Jansen, and Robertson (1993).
Round 2
Reviewer 2 Report
Comments and Suggestions for Authors
I appreciate the in- depth, thoughtful and rigorous responses to my review of this very interesting paper. Our Brazilian scientists are to be congratulated for conducting an important project that illustrates an often un realized area in vaccine uptake. The revisions are well placed and appropriate